# The Impact of Different Ventilatory Strategies on Clinical Outcomes in Patients with COVID-19 Pneumonia

**DOI:** 10.3390/jcm11102710

**Published:** 2022-05-11

**Authors:** Rihards P. Rocans, Agnese Ozolina, Denise Battaglini, Evita Bine, Janis V. Birnbaums, Anastasija Tsarevskaya, Sintija Udre, Marija Aleksejeva, Biruta Mamaja, Paolo Pelosi

**Affiliations:** 1Anesthesiology and Intensive Care Clinics, Riga East Clinical University Hospital, Hipokrata Street 2, LV-1079 Riga, Latvia; agnese.ozolina@icloud.com (A.O.); evitabine@gmail.com (E.B.); jvbirnbaums@gmail.com (J.V.B.); biruta.mamaja@aslimnica.lv (B.M.); 2Department of Anaesthesia and Intensive Care, Riga Stradiņš University, Dzirciema Street 16, LV-1007 Riga, Latvia; anastasija.carevska@gmail.com; 3Anesthesiology and Critical Care, San Martino Policlinico Hospital, IRCCS for Oncology and Neurosciences, 16132 Genoa, Italy; battaglini.denise@gmail.com (D.B.); ppelosi@hotmail.com (P.P.); 4Faculty of Medicine, Riga Stradiņš University, Dzirciema Street 16, LV-1007 Riga, Latvia; 5Faculty of Medicine, University of Latvia, Raina Boulevard 19, LV-1586 Riga, Latvia; sintija_udre@inbox.lv (S.U.); alekseyeva3@inbox.lv (M.A.); 6Department of Surgical Sciences and Integrated Diagnostics, University of Genoa, 16145 Genoa, Italy

**Keywords:** percutaneous tracheostomy, tracheostomy, COVID-19, intensive care

## Abstract

Introduction: The aim was to investigate the impact of different ventilator strategies (non-invasive ventilation (NIV); invasive MV with tracheal tube (TT) and with tracheostomy (TS) on outcomes (mortality and intensive care unit (ICU) length of stay) in patients with COVID-19. We also assessed the impact of timing of percutaneous tracheostomy and other risk factors on mortality. Methods: The retrospective cohort included 868 patients with severe COVID-19. Demographics, MV parameters and duration, and ICU mortality were collected. Results: MV was provided in 530 (61.1%) patients, divided into three groups: NIV (*n* = 139), TT (*n* = 313), and TS (*n* = 78). Prevalence of tracheostomy was 14.7%, and ICU mortality was 90.4%, 60.2%, and 30.2% in TT, TS, and NIV groups, respectively (*p* < 0.001). Tracheostomy increased the chances of survival and being discharged from ICU (OR 6.3, *p* < 0.001) despite prolonging ICU stay compared to the TT group (22.2 days vs. 10.7 days, *p* < 0.001) without differences in survival rates between early and late tracheostomy. Patients who only received invasive MV had higher odds of survival compared to those receiving NIV in ICU prior to invasive MV (OR 2.7, *p* = 0.001). The odds of death increased with age (OR 1.032, *p* < 0.001), obesity (1.58, *p* = 0.041), chronic renal disease (1.57, *p* = 0.019), sepsis (2.8, *p* < 0.001), acute kidney injury (1.7, *p* = 0.049), multiple organ dysfunction (3.2, *p* < 0.001), and ARDS (3.3, *p* < 0.001). Conclusions: Percutaneous tracheostomy compared to MV via TT significantly increased survival and the rate of discharge from ICU, without differences between early or late tracheostomy.

## 1. Introduction

The novel coronavirus severe acute respiratory distress syndrome (SARS-CoV-2) began its rapid spread in December 2019, causing a rapidly progressing respiratory disease (coronavirus disease-2019, COVID-19) with a high potential for evolving into multiple organ dysfunction (MODS) and death [1,2]. Severe COVID-19 often requires intensive care unit (ICU) admission, and it is associated with higher mortality and longer length of stay in ICU [3]. Patients with severe COVID-19 may require advanced respiratory support, including non-invasive ventilation (NIV) or invasive mechanical ventilation (MV) via a tracheal tube or a tracheostomy. Several studies have demonstrated the efficacy of NIV in moderate to severe COVID-19 [4]. However, vigorous spontaneous breathing efforts during NIV might be compatible with patient self-inflicted lung injury (P-SILI) development [5,6,7], with a two-fold higher risk of failure in COVID-19 than hypoxemic respiratory failure due to other etiologies [8]. Other studies have reported that NIV might not influence the outcome or natural course of COVID-19 disease [9]. Invasive MV via a tracheal tube appears to be the most common approach after failed NIV but is associated with higher all-cause mortality in COVID-19 patients [10,11,12]. Contrasting results have been published regarding early intubation in terms of survival benefits [13,14,15,16,17]. Tracheostomy is considered an effective alternative to invasive MV via tracheal tube since it helps to reduce dead-space ventilation and decreases airway resistance and the risk of aspiration, as well as the need for sedation, facilitating weaning from a ventilator [18]. During the COVID-19 pandemic, there was incomplete evidence that early tracheostomy is preferable to late tracheostomy, and there is no compelling evidence that percutaneous is preferable to surgical (open) [5,19,20,21]. Consequently, the appropriate MV strategy for COVID-19 respiratory failure is still disputed. Our hypothesis was that the early percutaneous tracheostomy approach may enhance the rate of survival and promote earlier successful discharge from ICU. The aim of the present study was to investigate the impact of different ventilator strategies, including NIV, invasive MV with tracheal tube (TT), and invasive MV with tracheostomy (TS), on ICU mortality and length of stay. Further, the impact of early vs. late percutaneous tracheostomy on survival and the risk factors associated with mortality was assessed in mechanically ventilated COVID-19 patients.

## 2. Materials and Methods

The study protocol and the informed consent form were approved by the Ethics Committee of Riga Stradins University (Approval Number 2-PEK-4/38/2022).

### 2.1. Patient Selection and Groups

This retrospective cohort study included 868 adult patients with severe COVID-19 who were treated in the ICU at Riga East University hospital from 1 January 2020 to 30 November 2021 (Figure 1). Data for patient selection were obtained from the statistics department according to a previously defined protocol and inclusion criteria. Given the observational nature of our study, all clinical management decisions were made by the attending physicians. Inclusion criteria: patients admitted to ICU with confirmed SARS-CoV-2 infection; patients with confirmed lung injury on computer tomography; requires any type of ventilation support (NIV, invasive MV via a tracheal tube or via tracheostomy). Exclusion criteria: patients with confirmed SARS-CoV-2 infection not associated with severe lung injury (polytrauma, malignancy, etc.). Based on inclusion criteria, 530 patients were selected for further analysis and divided into three groups according to their definitive approach to continuous ventilatory support: non-invasive ventilation (NIV, *n* = 139), invasive MV with tracheal tube (TT, *n* = 313), and invasive MV with tracheostomy (TS, *n* = 78).

### 2.2. Outcomes

The primary outcome was to investigate the impact of different ventilator strategies (NIV, TT, and TS) on outcomes (ICU mortality and length of stay). The secondary outcomes were the impact of timing of percutaneous tracheostomy associated with mortality and risk factors associated with mortality (other than ventilation approach and timing of airway management).

### 2.3. Data Collection

Demographic data, the severity of COVID-19 disease, comorbidities, type of ventilation support (NIV via face mask or helmet, MV via TT or MV via TS), duration of ventilation, length of stay in the ICU, discharge and case-fatality rates from ICU, duration of hospitalization, as well as data on complications and course of the disease were manually precisely obtained from electronic health records. Detailed parameters of mechanical ventilation and severity scores such as SOFA and APACHE II were not included in our analysis due to retrospective study design without available data from electronic data sets.

### 2.4. Definitions

Early tracheostomy was defined as being performed ≤ 7 days after intubation [22]. Non-invasive ventilation was defined as respiratory support delivered via face mask or helmet using BiPAP mode. The severity of COVID-19 and complications were determined by clinicians using the following: Berlin definition of ARDS for mild, moderate, and severe syndrome [23]; Sepsis-3 definition for sepsis [24]; KDIGO criteria for acute kidney injury [25]. The WHO Living guidance was used to classify COVID-19 severity (mild, moderate, severe, and critically severe) [26]. ICU length of stay was the timing between admission to the ICU and discharge from the ICU to the ward. Hospital length of stay was the timing between admission to the hospital and discharge from the hospital to home. Mean ventilator-free days were defined as the time from the last use of any mechanical ventilation support to discharge from ICU. The mortality rate was defined as case fatality rates during the stay in ICU and case fatality rates in ward after discharge from ICU. Survival rate was defined as successful discharge from ICU to the ward. All tracheostomies were performed using percutaneous technique by teams of intensive care specialists and pulmonary physicians with more than 20 years of experience according to local hospital guidelines and individual clinical decisions made by a multidisciplinary team.

### 2.5. Statistical Analysis

Statistical analysis was performed using SPSS Statistics for Windows, Version 26.0. (IBM Corp. Armonk, NY, USA). No sample size calculation was performed due to the retrospective nature of this study. The Kolmogorov–Smirnov test was used to evaluate whether the datasets conformed to a normal distribution. Variables were presented as mean and 95% confidence interval (CI 95), median ± interquartile range (IQR), and proportions as appropriate. Differences in data distribution between the groups were evaluated using the Mann–Whitney U test or Kruskal–Wallis test for non-parametric datasets and the two-sample *t*-test or ANOVA for datasets conforming with normal distribution. The Chi-square test was applied for nominal variable sets. Binary logistic regression models were used to obtain odds ratios for risk factors associated with mortality. Kaplan–Meier Log Rank tests were performed to compare survival between groups. Statistical significance was assumed if two-tailed *p* < 0.05.

## 3. Results

### 3.1. Characteristics of the Overall COVID-19 Population

The patients’ inclusion flow is reported in Figure 1.

In total, 868 patients—448 (51.6%) men and 420 (48.4%) women—were included. The mean age was 63.0 (95% CI 62.1 to 63.9) years. The overall mortality rate for all COVID-19 patients in the ICU was 54.6% (*n* = 474). Patients who died in the ICU were older than those successfully discharged from the ICU (65.6 ‘64.5–66.8’ vs. 59.8 ‘58.4–61.2’, *p* < 0.001). Patients who did not receive mechanical ventilatory support had an ICU mortality rate of 30.2% (*n* = 102). Of the cases reviewed, 38.9% (*n* = 338) did not receive mechanical ventilation.

### 3.2. Characteristics of the COVID-19 Population According to the Type of Ventilatory Support

For further analysis, 530 consecutive patients with COVID-19 were included. Of those 26.2% (*n* = 139) were assigned to NIV, while 59.1% (*n* = 313) patients were allocated to invasive MV with TT and 14.7% (*n* = 78) to invasive MV with TS groups. Table 1 displays baseline demographic characteristics, comorbidities, as well as COVID-19 severity and complication rates for the groups and in the overall population with mechanical ventilation.

The TS group showed a higher rate of secondary bacterial pneumonia (55.1% vs. 39.0%; respectively, *p* = 0.045) when compared to TT group and acute respiratory distress syndrome as a complication was observed more often in TS compared to TT and NIV groups (41.0% vs. 29.4%; *p* = 0.024, 41.0% vs. 18%; respectively, *p* < 0.001). The prevalence of percutaneous tracheostomy was 9% of the overall population and 14.7% of patients who received invasive MV. As shown in Table 2, the duration of MV was longer in the TS group compared to the TT group (19.9 vs. 7.16 days, respectively, *p* < 0.001).

### 3.3. Impact of Different Ventilator Strategies on Mortality

The overall ICU mortality rate for patients requiring any type of mechanical ventilatory support was 70.2% (*n* = 372). As shown in Table 2, the NIV group had the lowest mortality rate in the ICU at 30.2%. In contrast, the TT group demonstrated the highest overall mortality rate of 90.4% in the ICU and a higher rate when compared to the TS group (60.2%), *p* < 0.001; even though the time from ICU admission to intubation in both groups was similar: 4.8 and 4.4 days; *p* = 0.365. Causes of death in the TS group were not related to the percutaneous tracheostomy procedure and its possible complications. Figure 2 shows statistically significant differences in survival between TS and TT groups, as well as between NIV and TT groups, *p* < 0.001. At multiple logistic regression, the TS group has higher odds of survival compared to NIV or TT (OR 6.3, *p* = 0.001), even after adjusting for age and the presence of ARDS. Patients who only receive invasive mechanical ventilation have higher odds of survival compared to those who receive NIV in ICU prior to relying on invasive ventilation (OR 2.7, *p* = 0.001).

### 3.4. Impact of Different Ventilator Strategies on ICU Length of Stay

The overall mean length of ICU stay was 12.5 (9.6–14.3), which was longer in TS compared to NIV and TT groups and similar between NIV and TT groups (22.2. ‘16.0–28.4’ vs. 10.5 ‘8.0–12.9’ and 10.7 ‘18.3–13.3’ days, respectively, *p* < 0.001). Despite the prolonged ICU stay, multiple logistic regression showed that percutaneous tracheostomy of an intubated patient increases the odds of discharge from the ICU (OR 6.3, *p* < 0.001) when adjusted for age and presence of ARDS.

### 3.5. Impact of the Timing of Airway Management on Mortality

The mean time for NIV was 1.5 days (1.4–1.7) after ICU admission. The timing or duration of NIV between survivors and patients who died in the ICU was not different. As shown in Table 3, the mean intubation time was not different for survivors and those who died in the ICU (15.6 days (12.1–19.1) and 12.4 days (10.3–14.6), respectively). Percutaneous tracheostomy was performed on average 9.6 days after intubation and 13.2 days (11.5) after admission to the ICU. The mean time from intubation to tracheostomy was 10.5 days (7.5–13.3) for survivors and 9.4 days (7.9–10.9) for patients who died in the ICU. Early percutaneous tracheostomy was performed in 46.2% of patients, while late tracheostomy in 53.8% of patients. The survival rate was comparable in patients who received early or late tracheostomy (44.4% vs. 38.1%, *p* = 0.101).

### 3.6. Risk Factors Associated with Mortality

Multiple logistic regression analysis of risk factors in 530 mechanically ventilated patients suggests that the odds of death in the ICU increase with age (OR 1.032, *p* < 0.001), chronic renal disease (OR 1.57, *p* = 0.019), and obesity (OR 1.58, *p* = 0.041). When adjusted for older age (OR 1.025, *p* < 0.001), multiple logistic regression analysis of complications reveals that the presence of sepsis (OR 2.8, *p* < 0.001), acute kidney injury (OR 1.7, *p* = 0.049), multiple organ dysfunction (OR 3.2, *p* < 0.001), acute respiratory distress syndrome (OR 3.3, *p* < 0.001) increase the odds of death in the ICU.

## 4. Discussion

In MV patients with COVID-19 pneumonia, we found that (1) ICU mortality was higher in the TT (90.4%) group as compared to TS (60.2%) and NIV groups (30.2%); (2) odds of survival and ICU discharge were increased in TS compared to TT group, but with a longer ICU length of stay; (3) timing of NIV, endotracheal intubation, and percutaneous tracheostomy did not significantly impact on ICU mortality; (4) the use of NIV before relying on invasive MV increased mortality; (5) other risk factors associated with mortality included age, obesity, chronic renal disease, obesity, development of complications such as sepsis, acute kidney injury, MODS, and ARDS.

Our retrospective cohort study included 530 adult patients with severe COVID-19 who received mechanical ventilatory support in the ICU. These are the first clinical observational data in Latvia, showing that the overall mortality rate for patients requiring any type of ventilatory support was higher (70.2%) than those reported in Germany (51%) and published in an international study (37%) [27,28]. These differences in mortality rates between ventilator strategies in ICU are at risk of confounding factors. According to recent findings, the need for a specific ventilator approach could be guided by the identification of COVID-19 phenotypes (1 or L and 2 or H) [29], taking into account the risk of aerosol generator procedures, as well as the development of patient self-inflicted lung injury (P-SILI) [30]. In another perspective, preemptive intubation was proposed to avoid progressive acute respiratory failure [31]. Recent data confirmed that in patients receiving NIV, late intubation was associated with worse clinical presentation and more severe disease than early, suggesting that the delay in intubation might have an impact on patient’s outcome [7]. However, our most unexpected finding was that patients who solely received invasive MV had a higher survival chance than those who received NIV in the ICU before relying on invasive ventilation (OR 2.7, *p* = 0.001), suggesting a possible role of NIV in promoting P-SILI in COVID-19 patients [30].

Patients with COVID-19 typically experience long periods of MV; thus, it is possible that tracheostomy may yield potential survival benefits, thus facilitating weaning from MV. Indeed, in our study, patients who received tracheostomy had a higher survival rate than those who were ventilated via endotracheal tube. These findings are consistent with data reported from previous studies [18,32,33]. The optimal timing for tracheostomy provided conflicting evidence [13,14,15,16,21,34,35]. In our study, longer ICU stay was found in the TS group. On the opposite, some previous studies in non-COVID-19 patients reported that early tracheostomy was associated with shorter overall ICU stay when compared to late tracheostomy. Patients who received invasive MV with TS had the longest ICU length of stay (22.2 days), followed by the invasive MV with TT (10.7 days) and then by NIV approach (10.5 days). Further, the longest hospital stay was observed in TS patients (49.7 days), followed by the patients with NIV (25.3 days) and TT (19.2 days) ventilatory strategies. These findings may be explained by the fact that the TS patients had better survival rates in ICU and required rehabilitation resulting in an extended hospital treatment period.

Considering age and presence of ARDS in the multiple logistic regression, we found that percutaneous tracheostomy increases the odds of being discharged from the ICU by 6.3 times. We found an increase in successful discharge rates from the ICU in the TS group, suggesting that tracheostomy may help to reduce dead-space ventilation, airway resistance, and the risk of aspiration, as well as the requirement for sedation and enhancing secretion clearance [18,32]. Some studies have outlined that COVID-19 causes myopathy of the diaphragm that is distinct from that of control-ICU patients, with a comparable duration of mechanical ventilation and ICU length of stay [36]. The decreased rate of myopathy of the diaphragm and P-SILI could explain the improved survival in patients receiving percutaneous tracheostomy, although this is yet to be tested in further studies. We also found that older age, obesity, and chronic renal disease were risk factors for mortality in the ICU, in line with previous findings [37,38]. Complications such as sepsis, acute kidney injury, MODS, and ARDS substantially increased the odds of death in the ICU, as reported in a recent meta-analysis [39]. Patients receiving invasive MV with percutaneous tracheostomy exhibited a higher probability of developing secondary bacterial pneumonia but without significantly affecting mortality rates. It is unclear whether secondary bacterial pneumonia developed before or after percutaneous tracheostomy, as subsequent bacterial pneumonia may have led the clinician to perform a percutaneous tracheostomy to improve airway secretion clearance.

### Limitations

This study has some significant limitations to be addressed. First, the cohort was drawn from a single cosmopolitan region, affecting the possible generalizability of the findings. Second, given the retrospective nature of our study, individual decision making was performed by the clinician and likely varied between cases. Third, knowledge and clinical management related to COVID-19 pneumonia are constantly advancing. Fourth, not all conceivable factors and conditions were taken into consideration. Factors such as prone positioning, airway pressures during mechanical ventilation and other parameters, the severity of lung damage as well as the severity of the disease (SOFA, APACHE II) could not be obtained due to the retrospective study design. Fifth, the follow-up was limited to the length of stay in the hospital; therefore, the long-term consequences are unknown. Sixth, further studies should be considered to fully understand the relation between mechanical ventilation and the mortality rate among COVID-19 patients, as well as the optimal timing of tracheostomy.

## 5. Conclusions

Mechanical ventilation with an endotracheal tube is frequently needed for patients with severe COVID-19 pneumonia. Our data suggest that early as well as late percutaneous tracheostomy may improve survival and discharge rate from ICU while increasing the length of stay in ICU compared to sustained invasive mechanical ventilation through an endotracheal tube. Nevertheless, our data suggest that ICU patients with severe COVID-19 pneumonia who require any type of invasive mechanical ventilation might be at a higher risk of mortality.

## Figures and Tables

**Figure 1 jcm-11-02710-f001:**
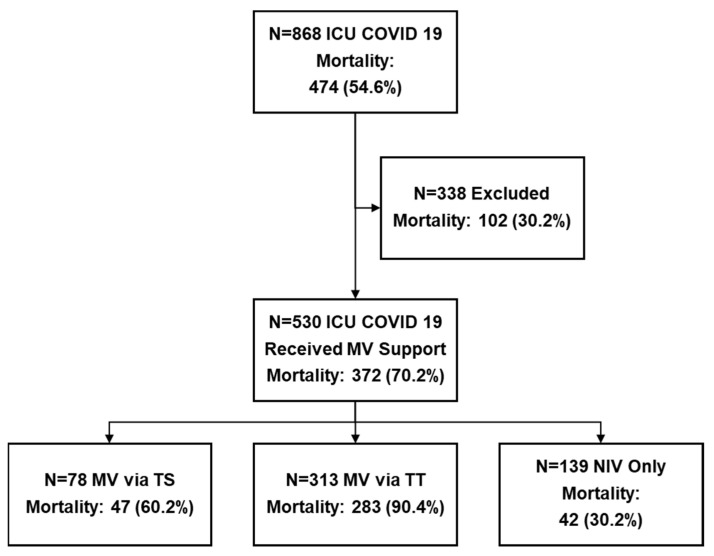
Patients’ inclusion flow. Abbreviations: COVID-19—coronavirus disease 2019; ICU—intensive care unit; MV—mechanical ventilations; TS—tracheostomy; TT—tracheal tube.

**Figure 2 jcm-11-02710-f002:**
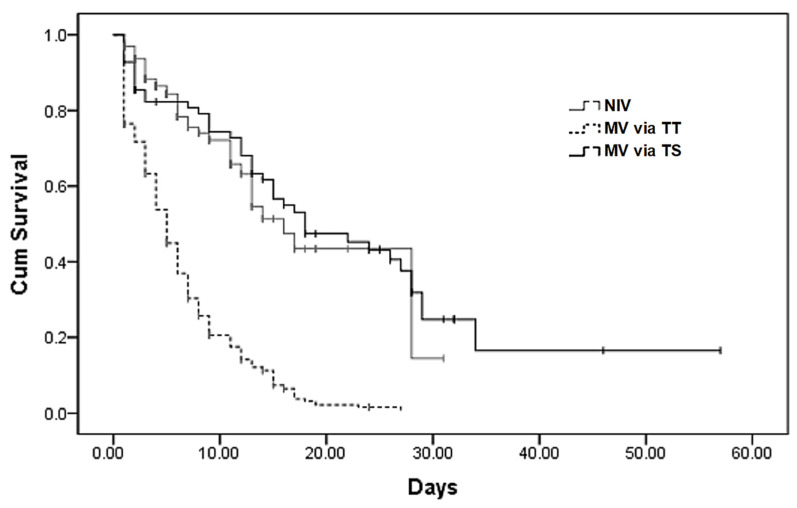
Kaplan–Meier survival plots for the approach of mechanical ventilation. Legend. Pairwise log rank comparisons of survival showed statistically significant differences in survival between TS and TT, as well as between NIV and TT, *p* < 0.001. No statistically significant log rank differences in survival were found between NIV and TS, *p* = 0.905. Abbreviations: NIV—non-invasive ventilation; MV—mechanical ventilation; TT—invasive mechanical ventilation with tracheal tube; TS—invasive mechanical ventilation with tracheostomy.

**Table 1 jcm-11-02710-t001:** Demographic characteristics, comorbidities, COVID-19 severity, and complications. Data are presented as mean and 95% confidence intervals, median interquartile ranges, or numbers and percentages as appropriate. Abbreviations: NIV (non-invasive ventilation); TT (invasive mechanical ventilation with tracheal tube); TS (invasive mechanical ventilation with tracheostomy); COPD (chronic obstructive pulmonary disease); ARDS (acute respiratory distress syndrome); BMI (body mass index); MODS (multiple organ dysfunction syndrome); PE (pulmonary embolism).

Ventilation Approach	Overall *n* = 530	NIV *n* = 139	TT *n* = 313	TS *n* = 78	*p*-Value
**Demographical data**					
Age, years	63.0 (62.1–63.1)	63.0 (61.0–65.4)	65.0 (63.5–66.5)	60.1 (57.5–62.8)	0.124
Women, *n* (%)	268 (50.6%)	79 (51.8%)	151 (48.2%)	38 (48.7%)	0.314
**Comorbidities**					
Coronary artery disease, *n* (%)	290 (54.7%)	84 (60.4%)	169 (54.0%)	37 (47.4%)	0.059
Diabetes mellitus, *n* (%)	150 (28.3%)	40 (28.8%)	91 (29.1%)	19 (24.4%)	0.691
COPD, *n* (%)	32 (6.0%)	6 (4.3%)	21 (6.7%)	5 (6.4%)	0.437
Malignancy, *n* (%)	151 (28.5%)	46 (33.1%)	82 (26.2%)	23 (29.5%)	0.391
Chronic renal disease, *n* (%)	71 (13.4%)	19 (13.7%)	48 (15.3%)	4 (5.1%)	0.164
Obesity (BMI > 30 kg/m^2^), *n* (%)	90 (17.0%)	25 (18.0%)	52 (16.6%)	13 (16.7%)	0.561
**COVID-19 severity**					
Mild, *n* (%)	17 (3.2%)	2 (1.4%)	12 (3.9%)	3 (4%)	0.803
Moderate, *n* (%)	86 (16.2%)	24 (17.3%)	53 (17.3)	9 (12%)	
Severe and Critical, *n* (%)	417 (78.7%)	113 (81.3%)	241 (78.8%)	63 (84.0%)	
**Complications**					
Sepsis, *n* (%)	188 (35.5%)	19(13.7%)	129 (41.2%) *^,^**	40 (51.3%) *	<0.001
Acute kidney injury, *n* (%)	94 (17.7%)	8 (5.8%)	68 (21.7%) *	18 (23.1%) *	<0.001
MODS, *n* (%)	158 (29.8%)	8 (5.8%)	121 (38.7%) *	29 (37.2%) *	<0.001
PE, *n* (%)	47 (8.8%)	10 (7.2%)	33 (10.5%)	4 (5.1%)	0.804
Stroke, *n* (%)	7 (1.3%)	0 (0)	4 (1.3%)	3 (3.86)	0.058
Secondary bacterial pneumonia, *n* (%)	224 (42.3%)	59 (42.4%)	122 (39.0%)	43 (55.1%)	0.107
ARDS, *n* (%)	149 (28.1%)	25 (18.0%)	92 (29.4%) *	32 (41.0%) *	<0.001

The * symbol is used to mark statistical significance when compared only to NIV group. The ** symbol is used to mark statistical significance when compared only to TS group.

**Table 2 jcm-11-02710-t002:** Impact of airway and ventilation approach on survival rates in terms of successful discharge from ICU. Data are presented as mean and 95% confidence intervals, median interquartile ranges, or numbers and percentages as appropriate. Abbreviations: NIV (non-invasive ventilation); TT (invasive mechanical ventilation with tracheal tube); TS (invasive mechanical ventilation with tracheostomy); ICU (intensive care unit).

Ventilation Approach	Overall *n* = 530	NIV *n* = 139	TT *n* = 313	TS *n* = 78	*p*-Value
**Mean days since first symptoms of COVID-19 at hospital admission**	8.2 (7.1–9.3)	8.2 (7.3–9.1)	8.1 (7.4–8.7)	9.0 (6.6–11.3)	0.489
**Mean duration of NIV, days**	5.6 (4.2–7.1)	8.58 (7.4–9.61)	4.8 (4.1–5.4) *	4.4 (3.1–5.7) *	<0.001
**Use of NIV before relying on invasive MV, *n* (%)**		–	194 (62%)	41 (52.6%)	0.155
**Time from admission in ICU to intubation, days**	4.2 (3.6–4.9)	–	4.3 (3.9–4.8)	4.0 (3.0–5.0)	0.365
**Duration of invasive mechanical ventilation, days**	9.7 (6.7–12.8)	–	7.16 (6.4–7.9) **	19.9 (16.1–23.7)	<0.001
**Mean length of stay in ICU, days**	12.3 (8.4–16.2)	10.5 (8.0–12.9) **	10.7 (8.3–13.3) **	22.2 (16.0–28.4)	<0.001
**Mean ventilator-free days**	6.9 (5.7–8.1)	6.1 (3.4–8.9)	6.9 (5.8–8.0)	7.1 (5.0–9.2)	0.240
**Discharged from the ICU, *n* (%)**	158 (29.8%)	97 (69.8%) **	30 (9.6%) *^,^**	31 (39.7%)	<0.001
**Mortality rates in the ICU, *n* (%)**	372 (70.2%)	42 (30.2%) **	283 (90.4%) *^,^**	47 (60.2%)	<0.001
**Mortality rates in Ward after discharge from ICU, *n* (%)**	18 (11.4%)	6 (6.2%) **	7 (23.3%) *^,^**	5 (16.12%)	0.022
**Duration of hospitalization, days**	25.4 (18.6–32.2)	25.3 (21.7–26.7) **	19.2 (12.5–24.9) *^,^**	49.7 (37.5–62.0)	<0.001

The * symbol is used to mark statistical significance when compared only to NIV group. The ** symbol is used to mark statistical significance when compared only to TS group.

**Table 3 jcm-11-02710-t003:** Comparison between survivors and non-survivors. Data are presented as mean and 95% confidence intervals, median interquartile ranges, or numbers and percentages as appropriate. NIV (non-invasive ventilation); TT (invasive mechanical ventilation with tracheal tube); TS (invasive mechanical ventilation with tracheostomy); ICU (intensive care unit); MV (mechanical ventilation); ARDS (acute respiratory distress syndrome); PE (pulmonary embolism); OR (odds ratio); BMI (body mass index); CI (confidence interval); COPD (chronic obstructive pulmonary disease).

	Survivors, *n* = 158	Non-Survivors, *n* = 372	*p*-Value	Univariate Analysis OR (*p*-Value; 95% CI)	Multivariate Analysis OR (*p*-Value; 95% CI)
**Age, years**	59.7 (58.4–61.2)	65.7 (64.5–66.9)	<0.001	1.03 (<0.001; 1.02–1.04)	1.03 (<0.001; 1.02–1.04)
**Females, *n* (%)**	83 (52.5%)	185 (49.7%)	0.223	-	-
**Comorbidities**					
Coronary artery disease, *n* (%)					
No	72 (45.6%)	168 (45.2%)			
Yes	86 (54.4%)	204 (54.8%)	0.503	-	-
Diabetes mellitus, *n* (%)					
No	115 (72.8%)	265 (71.2%)	0.437	-	-
Yes	43 (27.2%)	107 (28.8%)			
COPD, *n* (%)					
No	149 (94.3%)	349 (93.8%)	0.291	-	-
Yes	9 (5.7%)	23 (6.2%)			
Malignancy, *n* (%)					
No	108 (68.4%)	271 (72.8%)	0.295	-	*-*
Yes	50 (31.6%)	101 (27.2%)			
Chronic renal disease, *n* (%)					
No	145 (91.8%)	314 (84.4%)	0.025	2.06 (0.024; 1.09–3.88)	1.57 (0.04; 1.02–2.4)
Yes	13 (8.2%)	58 (15.6%)			
Obesity (BMI > 30 kg/m^2^), *n* (%)					
No	138 (84.8%)	292 (82.3%)	0.082	1.9 (0.01; 1.1–3.2)	1.58 (0.02; 1.08–2.3)
Yes	20 (15.2%)	80 (17.7%)			
**Complications**					
Sepsis, *n* (%)					
No	126 (79.7%)	216 (58.1%)	<0.001	2.8 (<0.001; 1.8–4.4)	2.8 (<0.001; 1.8–4.3)
Yes	32 (20.3%)	156 (41.9%)			
Acute kidney injury, *n* (%)					
No	145 (91.8%)	291 (78.2%)	<0.001	3.1 (<0.001; 1.7–5.8)	1.7 (0.049; 1.05–2.9)
Yes	13 (8.2%)	81 (21.8%)			
MODS, *n* (%)					
No	138 (87.3%)	234 (62.9%)	<0.001	4.1 (<0.001; 2.4–6.8)	3.15 (<0.001; 1.9–5.1)
Yes	20 (12.7%)	138 (37.1%)			
PE, *n* (%)					
No	147 (93.0%)	336 (90.3%)	0.404	*-*	-
Yes	11 (7.0%)	36 (9.7%)			
Stroke, *n* (%)					
No	155 (98.1%)	368 (98.9%)	0.431	*-*	*-*
Yes	3 (1.9%)	4 (1.1%)			
Secondary bacterial pneumonia, *n* (%)					
No	92 (58.2%)	214 (57.5%)	0.924	*-*	-
Yes	66 (41.8%)	158 (42.5%)			
ARDS, *n* (%)					
No	129 (81.6%)	252 (67.7%)	<0.001	2.1 (0.001; 1.3–3.3)	3.3 (<0.001; 2.1–5.1)
Yes	29 (18.4%)	120 (32.3%)			
**Ventilation approach:**					
**NIV, *n* (%)**	97 (69.8%)	42 (30.2%)	<0.001	12.5 (<0.001; 7.9–19.7)	-
**Invasive MV with TT, *n* (%)**	30 (9.6%)	283 (90.4%)	<0.001	0.07 (<0.001; 0.05–0.11)	
**Invasive MV with TS, *n* (%)**	31 (39.7%)	47 (60.2%)	<0.001	6.2 (<0.001; 3.5–11.2)	6.3 (<0.001; 3.4–11.6)
**Early, n (%) (*n* = 36)**	16 (44.4%)	20 (56.6%)	0.542		
**Late, n (%) (*n* = 42)**	16 (38.1%)	26 (61.9%)			
**Mean days since first symptoms of COVID-19 at hospital admission**	8.7 (7.4–9.3)	8.1 (7.6–8.7)	0.091	-	-
**Mean duration of NIV, days**	6.4 (5.1–8.8)	5.3 (4.7–7.0)	0.072	-	-
**Use of NIV prior to invasive MV, *n* (%)**	22 (49.2%)	213 (62.3%)	0.028	2.0 (0.02; 1.1–3.7)	2.7 (0.001; 1.5–4.1)
**Time from admission in ICU to intubation, days**	4.1 (3.6–4.6)	4.3 (3.8–4.9)	0.563	-	-
**Duration of invasive MV days**	16.2 (12.8–19.6)	8.5 (7.5–9.5)	<0.001	1.1 (<0.001; 1.03–1.1)	-
**Mean length of stay in ICU, days**	13.4 (11.8–15.1)	11.9 (8.7–13.3)	0.072	-	-
**Mean ventilator-free days**	7.6 (2.1–14.9)	1.7 (0.54–3.11)	0.064	-	-
**Duration of hospitalization, days**	20.9 (19.1–22.8)	14.8 (8.4–21.2)	0.276	-	-

## Data Availability

The datasets used and analysed during the current study are available from the corresponding author on reasonable request. The corresponding author will ensure that individual privacy is not compromised upon the transfer of datasets in case of any request.

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
