# Peer review of "The Impact of Different Ventilatory Strategies on Clinical Outcomes in Patients with COVID-19 Pneumonia"

_jcm, 2022, doi:10.3390/jcm11102710_

Round 1

Reviewer 1 Report

General Comments

This large retrospective observational study provides additional evidence regarding the impact of mechanical ventilation and artificial airway practices on outcomes of COVID-19 subjects with varying degrees of acute respiratory failure. As such I think it is potentially a valuable contribution to the medical literature on this topic. Unfortunately, there are number of apparent errors (incongruities in data) and vagaries in wording/definitions (or lack thereof) in manuscript that must be addressed and accounted for prior to reaching the threshold for publication acceptance. I think all of my concerns can be addressed with only a modest amount of additional effort in data presentation and writing.

These concerns are listed below.

Specific Comments/Concerns

Methods: Definitions (Line 102). NIV should be broken down into it components of mask/nasal CPAP vs. NIPPV and also the highest level of support needed (ie. CPAP level and PIP/PEEP for NIPPV).

COVID Severity: mild, moderate and severe are not specifically defined in terms of variables and parameters used to describe them expect for “ARDS”. The Berlin definition of ARDS also includes the same subcategories. The reviewer/reader should not have to ponder over these ambiguities. If the Berlin definition is what was meant, then further information is required as to the breakdown between mild, moderate and severe. Moreover, the criteria used to distinguish between COVID-19 “mild” and “moderate” versus Berlin definitions of “mild” and “moderate”.

It is not acceptable methodological practice to leave the reviewer or reader wondering what rules the investigators used to classify their subjects, particularly as this has an enormous impact on how to interpret the validity of their primary outcomes.

The same issue applies to the criteria used to define the presence of sepsis (eg. 3rd International Consensus definitions?) and whatever criteria/thresholds were used to define “acute renal damage” for which the long-accepted terminology is “acute kidney injury” (eg. RIFLE, AKIN, etc).

Results:

Figure 1 Flow Chart (Page 4): This is a little unorthodox in its layout and very confusing. But most importantly the data in the figure directly contradicts the mortality information and N among subgroups listed at the bottom of Table 2. This includes the juxtaposing subjects initially categorized as receiving only NIV (313) vs. TT (139) in Figure 1 verses Tables 1 and 2 in which those receiving only NIV (139) and TT (313). And then there is the problem of overall mortality rate of 54.6% (N=474) “for all COVID-19 patients in the ICU” (Lines 132-133).

The mortality discrepancy is caused by the misalignment in rows that appears to be caused during copyediting and is not the fault of the authors. Unfortunately, it took me several hours of going over the data to realize I was looking at the wrong row. This needs to be corrected by the editorial office.

Table 2, Row 4 vs. Row 6: The investigators need to explain how the “duration” of MV days can be statistically significant (P<0.001) between TT and TS by a mean difference of 12.7 days greater in the MV-TS group and yet “mean ventilator-free days” (actually mistakenly expressed as median and IQR) is only 1.7 days for MV-TT and is higher in the MV-TS (10.5 days) but is not significant (P=0.24). That cannot be true unless there was a methodological decision in how subjects who died prior to reaching unassisted breathing were accounted for (ie: any deaths prior to day 28 without achieving a period of > 48hr UAB would be. AS the mortality was 90.4% in the MV-TT group, that likely is the source.

More to the point, this problem stems from the fact that the authors did not define how ventilator-free days was calculated in their methodology. This is not acceptable and the discrepancy must be accounted for in order for, along with an explicit description of how ventilator-free days were calculated (eg. Day 28-MV days) and any caveats used in that determination (eg. must require > 48hrs without reinstitution of positive pressure ventilation, etc.) Again, the reviewers and the readers should not have to spend time trying to figure out the source of puzzling data like this. It should be readily apparent from reading a well-constructed and explicit methods section.

Finally, wherein asterisks were used to signify statistically significant differences between subgroups in Table 1, that practice does not appear at all in comparisons made in Table 2. This should be a uniform practice across data presentations.

Line 149: Should change wording to “were significantly different between all groups”. However, while that appears to apply for sepsis, the use of asterisks and the original wording “among groups” appears contradictory when applied to MODS and AKI where the asterisks are used more discreetly and matches the lack of apparent difference between TT and TS groupings. I’m assuming then that the asterisks used for differentiating ARDS (NIV and MV-TS) also means that the salient differences between TT and TS (29% vs. 41%) was not significant? Again, the reader and reviewers should not have to ponder what the investigators are trying to convey. I would strongly recommend that each significant comparison between each subgroup be identified individually by different symbols if necessary. In other words use one subgroup as the referent for all other comparisons in the figure legend (eg. *P<0.001 compared to NIV)

Discussion: The discussion section can probably be reduced by ~20-25% with some work on content and sentence structure. I estimate it is ~720 words so finding away to cover most of the topics with a reduction of ~150 words would help clear some space to discuss more pertinent issues regarding your study. For example, the hypothesis of Type-L vs. Type-H has lost much of its appeal as it appears more likely that the evolution of lung injury (at least with the earliest form and initial variants of SARS CoV-2) was slower (see Kallet, RH Respiratory Care 2021 “COVID-19 Year in Review” which is open access). I would target this discussion first as a way to reduce the length of the discussion.

I think some of the limitations regarding the amount of data and the ability to capture it should be discussed briefly in the methodology (1-2 sentences). Since duration of MV and the role of tracheostomy in weaning depends upon severity of oxygen transfer dysfunction, elevated Vd/Vt and reduced chest compliance (especially in those with morbid obesity) not being able to access that data (to the degree it was even collected) greatly limits what can be argued about the relevance of the favorable data associated with the MV-tracheostomy group. In addition, SOFA, APACHE or SAPS data might have helped explain the extremely high mortality in the MV-TT subgroup which I only recall seeing in the initial reports from China, Italy, New York City), The central tendency for mortality reported in 2020 was ~50%). It might be helpful to review your data to see whether mortality was skewed towards 2020 vs. 2021 and whether resource issues during the early surges in Latvia might have played a role. Personally, I would find this much more interesting than devoting more than passing mention of Type-L, Type-H “phenotypes” (which I consider to be a misnomer).  

Conclusions: I’m a little uneasy about the statement “those who require ARE at higher risk of mortality REGARDLESS of their features or comorbidities”. The investigators were unable to accrue sufficiently detailed data to make such a claim (as I discussed above). It would be much better to word this more conservatively (eg. “our data suggests that …. may be more at risk)

Author Response

Dear Reviewer,

Thank you for taking the time and effort to review the manuscript. We appreciate your valuable comments and suggestions, which helped us improve the quality of the manuscript.

All changes in the latest manuscript are marked with the ‘’track changes’’ option and referenced in the answers we have provided. Please find the latest manuscript in the attachments.

Please find our answers to all your comments below.

Kind regards,

The Authors

 J Clin Med-Rocans et al MV strategies-COVID-19

General Comments

This large retrospective observational study provides additional evidence regarding the impact of mechanical ventilation and artificial airway practices on outcomes of COVID-19 subjects with varying degrees of acute respiratory failure. As such I think it is potentially a valuable contribution to the medical literature on this topic. Unfortunately, there are number of apparent errors (incongruities in data) and vagaries in wording/definitions (or lack thereof) in manuscript that must be addressed and accounted for prior to reaching the threshold for publication acceptance. I think all of my concerns can be addressed with only a modest amount of additional effort in data presentation and writing.

Answer: we thank the Reviewer for the positive comments.

These concerns are listed below.

Specific Comments/Concerns

Methods: Definitions (Line 102). NIV should be broken down into it components of mask/nasal CPAP vs. NIPPV and also the highest level of support needed (ie. CPAP level and PIP/PEEP for NIPPV).

Answer: We thank the reviewer for this comment. Unfortunately, detailed information of ventilation parameters used for NIV is not available in our electronic records, but in the revised manuscript we better specified that only NIV via face mask or helmet interface using BiPAP mode were applied in our cohort of patients (lines 109-110). We recognize the lack of information on ventilation pressures as a limitation that is relevant not only to NIV but to all types of ventilation. This additional issue was better addressed in line 315.

COVID Severity: mild, moderate and severe are not specifically defined in terms of variables and parameters used to describe them expect for “ARDS”. The Berlin definition of ARDS also includes the same subcategories. The reviewer/reader should not have to ponder over these ambiguities. If the Berlin definition is what was meant, then further information is required as to the breakdown between mild, moderate and severe. Moreover, the criteria used to distinguish between COVID-19 “mild” and “moderate” versus Berlin definitions of “mild” and “moderate”.

It is not acceptable methodological practice to leave the reviewer or reader wondering what rules the investigators used to classify their subjects, particularly as this has an enormous impact on how to interpret the validity of their primary outcomes.

Answer: Thank you for this comment. We agree that more clarity about definitions is needed in the methodology. To avoid ambiguities, in the revised version of the manuscript we added new references for following definitions: ARDS, Sepsis, acute kidney injury and COVID-19 severity in lines 111-113.

The same issue applies to the criteria used to define the presence of sepsis (eg. 3rd International Consensus definitions?) and whatever criteria/thresholds were used to define “acute renal damage” for which the long-accepted terminology is “acute kidney injury” (eg. RIFLE, AKIN, etc).

Answer: Thank you for highlighting this important point. The term acute renal damage has been modified to “acute kidney injury” throughout the revised manuscript. As mentioned before, we included following statements - The presence of sepsis was determined using the Sepsis-3 definition. The presence of acute kidney injury was determined using KDIGO criteria in lines 111-113.

Results:

Figure 1 Flow Chart (Page 4): This is a little unorthodox in its layout and very confusing. But most importantly the data in the figure directly contradicts the mortality information and N among subgroups listed at the bottom of Table 2. This includes the juxtaposing subjects initially categorized as receiving only NIV (313) vs. TT (139) in Figure 1 verses Tables 1 and 2 in which those receiving only NIV (139) and TT (313). And then there is the problem of overall mortality rate of 54.6% (N=474) “for all COVID-19 patients in the ICU” (Lines 132-133).

The mortality discrepancy is caused by the misalignment in rows that appears to be caused during copyediting and is not the fault of the authors. Unfortunately, it took me several hours of going over the data to realize I was looking at the wrong row. This needs to be corrected by the editorial office.

Answer: Thank you for noting this issue. In the revised version of the manuscript, we have made corrections to Figure 1 according to your suggestions. We sincerely apologize that you had inconveniences caused by the misalignment during copyediting and we modified accordingly.

Table 2, Row 4 vs. Row 6: The investigators need to explain how the “duration” of MV days can be statistically significant (P<0.001) between TT and TS by a mean difference of 12.7 days greater in the MV-TS group and yet “mean ventilator-free days” (actually mistakenly expressed as median and IQR) is only 1.7 days for MV-TT and is higher in the MV-TS (10.5 days) but is not significant (P=0.24). That cannot be true unless there was a methodological decision in how subjects who died prior to reaching unassisted breathing were accounted for (ie: any deaths prior to day 28 without achieving a period of > 48hr UAB would be. AS the mortality was 90.4% in the MV-TT group, that likely is the source.

More to the point, this problem stems from the fact that the authors did not define how ventilator-free days was calculated in their methodology. This is not acceptable and the discrepancy must be accounted for in order for, along with an explicit description of how ventilator-free days were calculated (eg. Day 28-MV days) and any caveats used in that determination (eg. must require > 48hrs without reinstitution of positive pressure ventilation, etc.) Again, the reviewers and the readers should not have to spend time trying to figure out the source of puzzling data like this. It should be readily apparent from reading a well-constructed and explicit methods section.

Answer: Thank you for the careful assessment of data. We investigated the issue regarding mean ventilator free days and found that there has been a mistake when copying mean values and confidence intervals from statistics interface. The correct values have been added in Table 2 of the revised manuscript. We are relieved you found this error and regret that it caused confusion. The p value of 0.24 is unchanged since it was copied correctly. We added the definition of mean ventilator free days in lines 116-118.

Finally, wherein asterisks were used to signify statistically significant differences between subgroups in Table 1, that practice does not appear at all in comparisons made in Table 2. This should be a uniform practice across data presentations.

Answer: Thank you for your suggestion. We made appropriate asterisk in Table 1 and 2 by adding sentences - The * symbol is used to mark statistical significance when compared only to NIV group. The ** symbol is used to mark statistical significance when compared only to TS group as seen in lines 166,167 and lines 180,181.

Line 149: Should change wording to “were significantly different between all groups”. However, while that appears to apply for sepsis, the use of asterisks and the original wording “among groups” appears contradictory when applied to MODS and AKI where the asterisks are used more discreetly and matches the lack of apparent difference between TT and TS groupings. I’m assuming then that the asterisks used for differentiating ARDS (NIV and MV-TS) also means that the salient differences between TT and TS (29% vs. 41%) was not significant? Again, the reader and reviewers should not have to ponder what the investigators are trying to convey. I would strongly recommend that each significant comparison between each subgroup be identified individually by different symbols if necessary. In other words use one subgroup as the referent for all other comparisons in the figure legend (eg. *P<0.001 compared to NIV)

Answer: Thank you for this comment. We appreciate your careful check of our data. According to your suggestion, we made comparisons between three analysed groups clearer by adding asterisk in the Table 1 and 2 (lines 166,167 and lines 180,181).

Discussion: The discussion section can probably be reduced by ~20-25% with some work on content and sentence structure. I estimate it is ~720 words so finding away to cover most of the topics with a reduction of ~150 words would help clear some space to discuss more pertinent issues regarding your study. For example, the hypothesis of Type-L vs. Type-H has lost much of its appeal as it appears more likely that the evolution of lung injury (at least with the earliest form and initial variants of SARS CoV-2) was slower (see Kallet, RH Respiratory Care 2021 “COVID-19 Year in Review” which is open access). I would target this discussion first as a way to reduce the length of the discussion.

Answer: Thank you for this suggestion. We have provided the changes suggested in the discussion by adding and discussing the reference of Kallet and shortening the part regarding COVID-19 phenotypes. We have also shortened other parts of the discussion section accordingly (see lines 248-249, 254-263, 274-281).

I think some of the limitations regarding the amount of data and the ability to capture it should be discussed briefly in the methodology (1-2 sentences). Since duration of MV and the role of tracheostomy in weaning depends upon severity of oxygen transfer dysfunction, elevated Vd/Vt and reduced chest compliance (especially in those with morbid obesity) not being able to access that data (to the degree it was even collected) greatly limits what can be argued about the relevance of the favorable data associated with the MV-tracheostomy group. In addition, SOFA, APACHE or SAPS data might have helped explain the extremely high mortality in the MV-TT subgroup which I only recall seeing in the initial reports from China, Italy, New York City).

Answer: Thank you for this suggestion. In the revised version of the manuscript, we have added a sentence regarding data collection in the Methods section (lines 103-105) and additionally, we have added a limitations statement regarding disease severity scores (SOFA, APACHE) in both the Methods section (lines 103-105) and Discussion (lines 315-318).

 The central tendency for mortality reported in 2020 was ~50%). It might be helpful to review your data to see whether mortality was skewed towards 2020 vs. 2021 and whether resource issues during the early surges in Latvia might have played a role. Personally, I would find this much more interesting than devoting more than passing mention of Type-L, Type-H “phenotypes” (which I consider to be a misnomer).

Answer: We understand that the Reviewer finds this information more interesting. Centre of Disease Prevention and Control of Latvia had reported the overall incidence of COVID-19 mortality in 2021, but there is not previously published any scientific research data on COVID-19 mortality in Latvia. Therefore, our study presently is the first scientific demonstration on COVID-19 mortality rates in intensive care patients in Latvia. We would feel grateful if we could avoid of giving any statistical data reported by Centre of Disease Prevention and Control of Latvia as mortality rate might be influenced by unclear methodology as well as how the COVID-19 mortality was defined and detected.

Conclusions: I’m a little uneasy about the statement “those who require ARE at higher risk of mortality REGARDLESS of their features or comorbidities”. The investigators were unable to accrue sufficiently detailed data to make such a claim (as I discussed above). It would be much better to word this more conservatively (eg. “our data suggests that …. may be more at risk)

Answer: We rephrased our concluding statement (lines 328-330) in accordance with your suggestions.

Suggested Re-formatting of Fig 1 Flow chart (mortality information was placed in this example only because the information was present in one subgroup in the authors version)

Answer: Thank you for this comment. We have modified our Figure 1 Flow chart according to your suggestions.

Reviewer 2 Report

The authors Rocans et al describe the impact of different ventilatory strategies on clinical outcomes in patients with COVID-19 pneumonia. However, several issues need to be addressed:

- In table 1, how were ‘mild’, ‘moderate’ and ‘severe’ COVID defined? Are these the ARDS severity criteria?

- Of those patients in the ‘NIV’ group, what proportion received CPAP alone, and what proportion received BiPAP NIV? This data should be included.

- Of the TS and TT populations, what proportion of patients received NIV prior to intubation, and of those, how long did they receive NIV for? This data should be included. Were all patients started on NIV, then intubated if NIV alone was insufficient to maintain oxygenation?

- The P/F ratios and APACHE II scores on admission to ICU should be included for all patients – this would help explain why the attending physician decided on the initial modality of ventilation used.

- The statement “Patients who only receive invasive mechanical ventilation have higher odds of survival compared to those who receive NIV in ICU prior to relying on invasive ventilation (OR 2.7, p=0.001)” is not supported by data in any table or figure, but it makes up a key conclusion of the study. The authors have not shown any data for patients in the group who receive NIV in ICU prior to relying on invasive ventilation – this must be made clear in order to support their statement.

- In table 1, of the total patient population – if 78.7% had severe COVID, why do only 28.1% have ARDS as a complication? Surely those with ‘severe’ COVID must have had ARDS? The same question also applies for the NIV, TT and TS populations.

- In table 1, an asterix (*) is put next to several of the numerical values, however the meaning of the asterix is not explained anywhere.

Author Response

Dear Reviewer,

Thank you for taking the time and effort to review the manuscript. We appreciate your valuable comments and suggestions, which helped us improve the quality of the manuscript.

All changes in the latest manuscript are marked with the ‘’track changes’’ option and referenced in the answers we have provided. Please find the latest manuscript in the attachments.

Please find our answers to all your comments below.

Kind regards,

The Authors

Comments and Suggestions for Authors

The authors Rocans et al describe the impact of different ventilatory strategies on clinical outcomes in patients with COVID-19 pneumonia.

Answer: We thank the reviewer for this summary

However, several issues need to be addressed:

- In table 1, how were ‘mild’, ‘moderate’ and ‘severe’ COVID defined? Are these the ARDS severity criteria?

Answer: Thank you for this suggestion. This comment has been raised also by another reviewer. We agree with you that the definitions should be improved. In the revised manuscript, we have better clarified our statements regarding COVID-19 severity definitions in Methods (lines 113-114) and Table 1.

- Of those patients in the ‘NIV’ group, what proportion received CPAP alone, and what proportion received BiPAP NIV? This data should be included.

Answer: Thank you for raising this point. In our intensive care units, we applied NIV via face mask or helmet only using BiPAP mode. We added this information in lines 109-110.  We recognize the lack of information on ventilation modes and pressures as a general limitation as it could be important also regarding to invasive mechanical ventilation in methods section (lines 103-105) and in limitations (lines 315-318).

- Of the TS and TT populations, what proportion of patients received NIV prior to intubation, and of those, how long did they receive NIV for? This data should be included. Were all patients started on NIV, then intubated if NIV alone was insufficient to maintain oxygenation?

Answer: The duration of NIV prior to invasive MV is reported in the second row of Table 2. A portion of the patients were intubated shortly after admission to ICU and therefore they did not receive NIV in ICU prior to invasive MV. In the revised manuscript, we calculated the proportion of patients receiving NIV prior to invasive MV and added it in the third row of Table 2.

- The P/F ratios and APACHE II scores on admission to ICU should be included for all patients – this would help explain why the attending physician decided on the initial modality of ventilation used.

Answer: Thank you for this comment. We agree with the reviewer that the APACHE II score should be included for all patients at ICU admission. Unfortunately, in our clinical practice APACHE II score is not calculated daily and saved to electronical datasets. Therefore, such information was not available due to retrospective design of the study. We have added a limitation statement regarding disease severity scores (SOFA, APACHE II) in both the methods section (lines 103-105) and discussion (lines 316-328). As well as due to our study design, specific P/F values are not available beyond information on the presence/absence of ARDS.

- The statement “Patients who only receive invasive mechanical ventilation have higher odds of survival compared to those who receive NIV in ICU prior to relying on invasive ventilation (OR 2.7, p=0.001)” is not supported by data in any table or figure, but it makes up a key conclusion of the study. The authors have not shown any data for patients in the group who receive NIV in ICU prior to relying on invasive ventilation – this must be made clear in order to support their statement.

Answer: Thank you for reporting this suggestion. We agree with your comment. The proportion of patients receiving NIV prior to invasive MV and added it in the third row of Table 2. Comparison and odds ratios in use of NIV prior to invasive MV between survivors and non-survivors can now be found in Table 3 of the revised manuscript.

- In table 1, of the total patient population – if 78.7% had severe COVID, why do only 28.1% have ARDS as a complication? Surely those with ‘severe’ COVID must have had ARDS? The same question also applies for the NIV, TT and TS populations.

Answer: Thank you for clarifying this point. As per the nature of our electronic health records, Severe and Critically severe COVID-19 are inherently grouped together. According to WHO Living classification only Critically severe disease includes presence of ARDS. This explains the discrepancy between the rates of Severe + Critically severe COVID and rates of ARDS.

- In table 1, an asterix (*) is put next to several of the numerical values, however the meaning of the asterix is not explained anywhere.

Answer: We agree with the Reviewer that the significance of the asterixis should be better explained. In the revised manuscript, we have better clarified the use of asterisk in Table 1 and 2 by adding sentences - The * symbol is used to mark statistical significance when compared only to NIV group. The ** symbol is used to mark statistical significance when compared only to TS group as seen in l 166,167 and l180,181.

Submission Date

20 April 2022

Date of this review

23 Apr 2022 17:33:51

Round 2

Reviewer 1 Report

I'd like to thank the investigators for their hard work on this study and their diligence in complying with my concerns. I am satisfied with their revisions